# Preferentially Expressed Antigen in Melanoma Is a Multifaceted Cancer Testis Antigen with Diverse Roles as a Biomarker and Therapeutic Target

**Mukulika Bose** [1,2]

1   Department of Pediatric Oncology, Dana-Farber Cancer Institute, Boston, MA 02215, USA; mukulikabose90@gmail.com
2   Department of Pediatrics, Harvard Medical School, Boston, MA 02115, USA

**Abstract:** Preferentially expressed antigen in melanoma (PRAME) is a cancer testis antigen (CTA) that is selectively expressed in certain somatic tissues, predominantly in the testis, and is overexpressed in various cancers. PRAME family proteins are leucine-rich repeat proteins that are localized in the nucleus and cytoplasm, with multifaceted roles in immunity, during gametogenesis and in the overall reproduction process. It is a widely studied CTA and has been associated with the prognosis and therapeutic outcomes in patients with epithelial and non-epithelial tumors. PRAME has also been studied extensively as a therapeutic target. Moreover, it has been found to play a role in most of the well-known cancer hallmarks. Interestingly, the role of PRAME in tumorigenesis is paradoxical. Over the last decade, PRAME has garnered substantial interest as a target for immunotherapy. There are multiple clinical trials and pre-clinical studies targeting PRAME alone or in combination with other tumor antigens. This review article is an attempt to update our knowledge and understanding of the context-dependent oncogenic functions of PRAME in various carcinomas, and the current immunotherapeutic strategies, challenges, and perspectives on developing newer strategies to target PRAME for a better outcome.

**Keywords:** biomarker; cancer testis antigen; cancer hallmarks; PRAME; immunotherapy

## 1. Introduction

PRAME is a cancer/testis antigen (CTA) that stands for "preferentially expressed antigen in melanoma", also known as CT130 (cancer testis antigen 130), MAPE (melanoma antigen preferentially expressed in tumors), and OIP-4 (Opa-interacting protein 4). It was first characterized in 1997 as a tumor-associated antigen in cells isolated from a melanoma, and it encodes the epitope presented by the human leucocyte antigen (HLA)-A24 [1]. Later, it was identified in a yeast two-hybrid screen for proteins that bind outer membrane proteins of pathogenic bacteria [2].

PRAME belongs to the CTA gene family and encodes a membrane-bound protein recognized by T lymphocytes [3]. PRAME can be detected in many human malignancies, apart from its expression in the testes and limited expression in ovaries, adrenals, and endometrium. It is reported to be absent or have a low expression in most normal tissues; however, it is expressed not only in solid tumors, but also in leukemia cells [4]. Genome-wide demethylation in male germline cells leads to high expression of PRAME in the testes [1]. It has been reported that the PRAME gene is hypermethylated in normal tissues but hypomethylated in most malignant cells. The PRAME gene encodes a membrane-bound protein and causes autologous cytotoxic T-cell-mediated immune responses [1,5]. High levels of PRAME are found in different malignancies [6]. A recent systematic immunohistochemical study of >5800 epithelial and non-epithelial tumors conducted by the National Cancer Institute showed that PRAME was expressed in the testis and proliferative

endometrium among normal tissues [7]. A new study reported >50% of PRAME-positive lesions in a number of epithelial tumors [7].

Hanahan and Weinberg have clearly laid down the concept of cancer hallmarks in their milestone article, "Hallmarks of Cancer: The Next Generation" [8]. These hallmarks currently comprise ten capabilities acquired by cancer cells during the development of human tumors, including sustaining proliferative signaling, evading growth suppressors, resisting cell death, enabling replicative immortality, inducing angiogenesis, and activating invasion and metastasis, genome instability, inflammation, reprogramming of energy metabolism, and evading immune destruction [8]. PRAME plays a pivotal role in multiple cellular processes, including some of the cancer hallmarks. PRAME is not only an oncogenic molecule, but it also serves as an immunotherapeutic target. In recent years, there have been a lot of studies showing the oncogenic and immunogenic potential of PRAME in various cancer types. Despite clinical trials targeting PRAME, the lack of understanding on various aspects of PRAME biology poses the greatest challenge. There is a considerable amount of heterogeneity among different PRAME isotypes that has made it difficult to determine the specific function, clinical relevance, and suitability of PRAME as a therapeutic target in different cancers. Therefore, it is important to understand its expression patterns and molecular functions to accurately determine its potential use as a biomarker and therapeutic target. In this review article, we discuss the roles of PRAME in different cancer hallmarks, list its accountability as a biomarker, attempt to summarize therapeutic strategies that target PRAME in human cancers, and provide new perspectives on how novel approaches are required for better outcomes for PRAME-targeted therapies in the clinic.

## 2. Structure and Function of PRAME

### 2.1. Structure

The PRAME gene is located on chromosome 22 (22q11.22) [9] within the human immunoglobulin lambda gene locus [10], which contains a large number of VL gene segments that encode light chains during B cell development. This locus also contains several other non-immunoglobulin genes. The NCBI database annotates five PRAME mRNA transcripts ranging from 2.1–2.7 kb in length (2141, 2162, 2197, 2220, 2776 bases); the two shortest transcripts were the most abundantly expressed in testis and leukemia cell lines [11]. At least 17 different PRAME mRNAs have been reported, the largest of which is a 3329 base transcript that is clearly detectable in cancer cell lines such as K562, Hela, and HL60. Each of the major transcripts contains six exons, with four containing a coding sequence, and all encode an identical polypeptide of 509 amino acids and 15 distinct GT-AG introns [10]. There are 14 transcripts or splice variants of PRAME [12]. There are differences in the 5′ ends of these transcripts, indicating the existence of alternative transcription start sites. Moreover, the strong promoter activity in reporter assays is shown by the sequence around the proximal transcription start site, including exon 1a and the first intron of the PRAME gene (−165 to +365), thereby supporting the idea of alternative transcription start sites [11]. Out of the five validated PRAME transcripts, four contain unique 5′ untranslated regions (5′ UTRs), similar to that found in other primates. This suggests that these sequences may have a significant role in regulating PRAME expression in response to metabolic or developmental signals. The duplication rate of cancer testis antigen genes suggests roles in chemo-sensing, reproduction, or immunity [13]. The computer-predicted structure of PRAME from AlphaFold project as found in the Protein Data Bank (PDB) is shown in Figure 1A, and the .pdb file used to create a ball-and-stick model using biorender.com is shown in Figure 1B. [14].

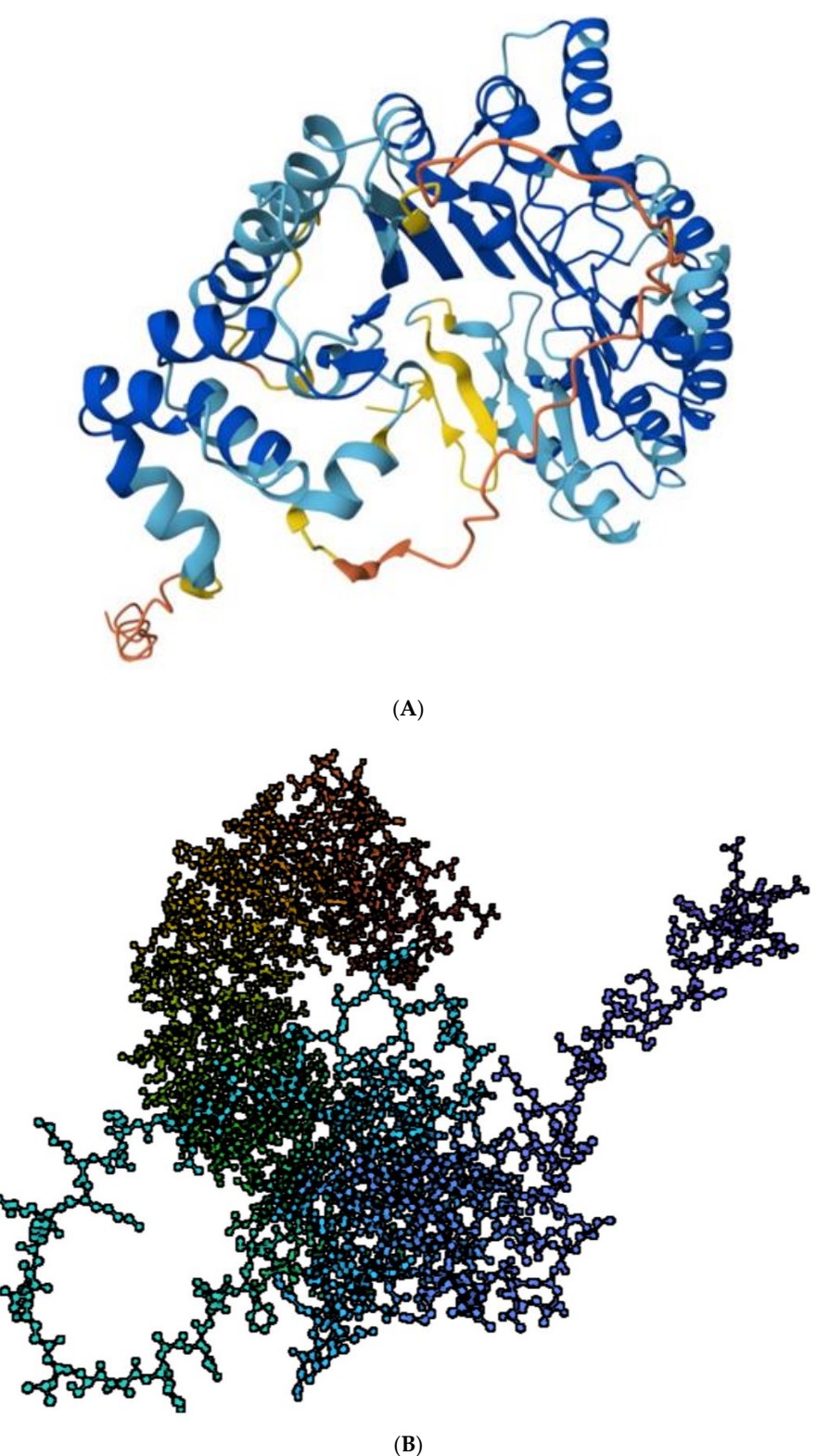

(**A**)

(**B**)

**Figure 1. (A) Structure of PRAME.** Predicted structure of PRAME (UniprotKB- P78395) adapted from Protein Data Bank (PDB) AlphaFold project, version 2. Color coding shows the model confidence: dark blue, very high (pLDDT > 90); light blue, confident (90 > pLDDT > 70): yellow, low (70 > pLDDT > 50); orange, very low (pLDDT < 50) https://www.rcsb.org/structure/af_afp78395f1 (accessed on 26 July 2023) [12,14]. (**B**) The computed structure of PRAME in a ball-and-stick model with color coding according to sequence as obtained from .pdb file using www.biorender.com (accessed on 26 July 2023).

## 2.2. Structure Determines Function

The PRAME protein consists of 509 amino acids, with a molecular mass of 57.89 kDa. Members of the PRAME family of proteins contain leucine-rich repeat (LRR) domains, sharing structural similarity with Toll-like receptors and localized in the nucleus and cytoplasm with various roles in germ cells [9,13,15]. The main functions of the PRAME family proteins include germline development, mainly the maintenance of embryonic stem cell pluripotency, development of primordial germ cells, and differentiation/proliferation of spermatogenic and oogenic cells [15]. These are also enriched in cytoplasmic organelles, such as rough endoplasmic reticulum, centrioles, Golgi vesicle, and germinal granules and are involved in the formation of the acrosome and sperm tail during spermiogenesis. The PRAME family genes remain transcriptionally active in the germline throughout the whole life cycle and are crucial for gametogenesis, with some members having specificity to either male or female germ cells, while others take part in the gametogenesis of both [15].

### 2.2.1. Differentiation

PRAME has been reported to function as a transcriptional repressor, inhibiting retinoic acid signaling through the retinoic acid receptors RARA, RARB, and RARG. It prevents retinoic-acid-induced cell-proliferation arrest, differentiation, and apoptosis (Figure 2). PRAME interacts with the RARA holoreceptor (via the ligand-binding domain) in a retinoic-acid-dependent way and recruits EZH2 to the promoter of RA target genes, thus repressing RAR signaling [16].

PRAME was first described as a dominant repressor of retinoic acid signaling, and being bound to the retinoic acid holoreceptor α (RARA), it recruits EZH2 to the promoter and acts as a co-repressor of RARE target genes, thus suppressing RA-induced differentiation, growth arrest, and apoptosis [16].

PRAME mRNA expression was found to increase with CML disease progression and was detected in late-chronic-phase CML patients before tyrosine kinase inhibitor (TKI) therapy. It was also associated with poorer therapeutic responses and ABL tyrosine kinase domain point mutations. PRAME protein expression inhibited granulocytic differentiation only in leukemia cell lines that differentiate along this lineage after all-trans retinoic acid (ATRA) treatment [17]. However, PRAME overexpression in normal hematopoietic progenitors inhibited myeloid differentiation in the presence and the absence of ATRA, and this phenotype was rescued when PRAME was knocked down in primary CML progenitors. This study suggested that PRAME expression causes inhibition of myeloid differentiation in certain myeloid leukemias, and that its function is lineage- and phenotype-dependent, in turn indicating that PRAME is a candidate for both prognostic and therapeutic applications [17].

### 2.2.2. Protein Degradation

PRAME is a chromatin-associated protein that is enriched at nuclear factor Y (NFY) target genes, in physical association with Elongin and Cullin-2 proteins [18]. It is the substrate-recognition component of a Cul2-RING (CRL2) E3 ubiquitin-protein ligase complex that mediates the ubiquitination (Figure 2) of truncated MSRB1/SEPX1 selenoproteins produced by failed UGA/Sec decoding for degradation [18,19]. The CRL2 (PRAME) complex is recruited to epigenetically and transcriptionally active promoter regions bound by nuclear transcription factor Y (NFY) and likely has a role in chromatin regulation [18]. PRAME contains a nuclear localization signal, with seven putative nuclear receptor (NR) boxes with the LXXLL consensus sequence enabling it to interact with nuclear receptors [20,21]. However, a large proportion of endogenous PRAME protein is observed in the cytoplasmic compartment in different cell lines [9,22].

### 2.2.3. Immune Target

In addition, PRAME also consists of HLA-specific epitopes, which are presented by MHC-class I molecules to the CD8+ T cells, thus eliciting an immune response against

PRAME+ tumors [23,24] (Figure 2). Downregulation and deletion of PRAME has been reported to be correlated with decreased immune cell infiltration and a cold tumor microenvironment in various cancers [24,25]. The immunomodulatory roles of PRAME have been discussed in this review in a later section.

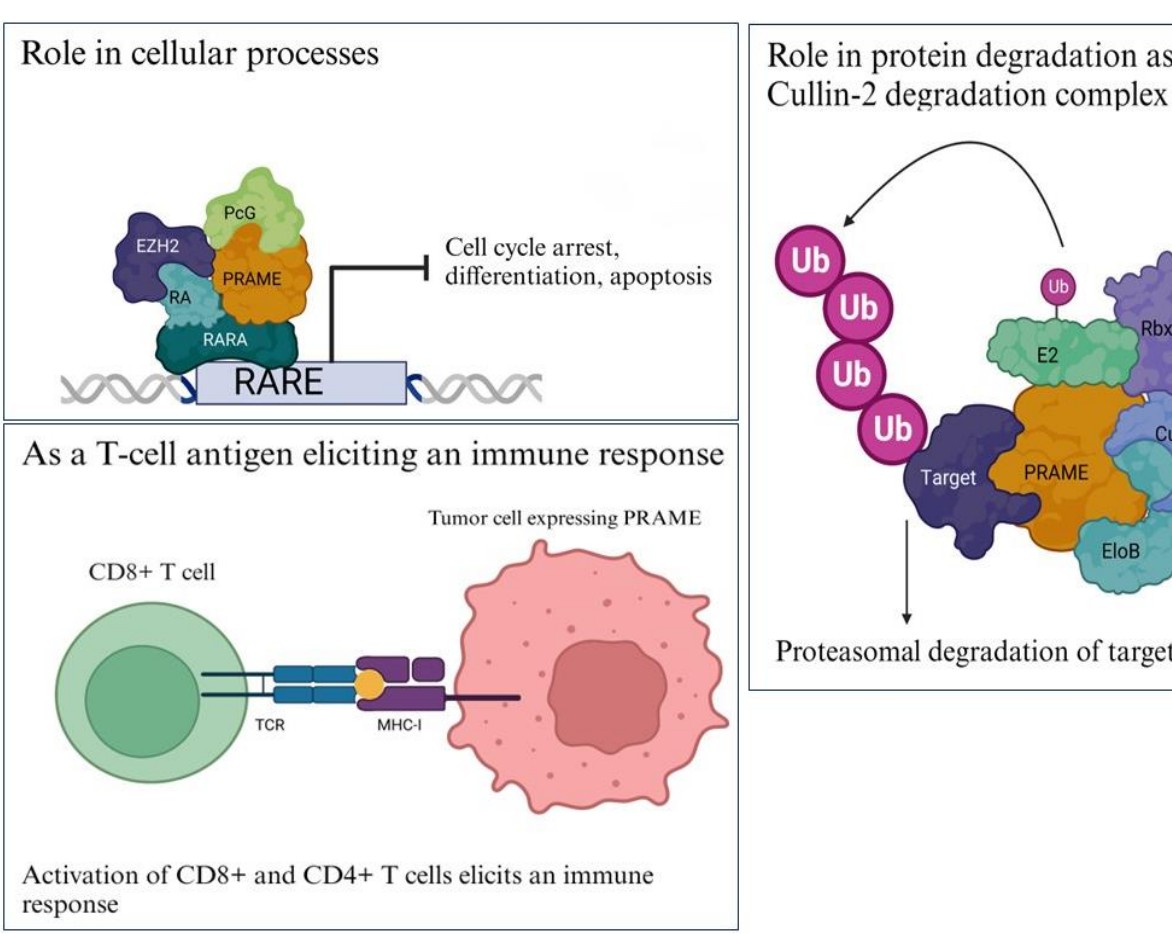

**Figure 2. Functions of PRAME.** (**Top left**). PRAME binds to RARα holoreceptor and acts as a corepressor to block transcription of RARE target genes involved in cell-cycle arrest, differentiation, and apoptosis; (**right**) PRAME is a part of the Cullin2-based E3 ligase complex that binds to its target proteins (shown in dark purple) and ubiquitinates them for proteasomal degradation; (**bottom left**) PRAME has HLA-specific epitopes that are presented by MHC class I and II molecules to activate CD8+ and CD4+ T cells, thus eliciting an immune response. Images have been adapted from various articles in the literature [16,18,23,24,26–29]. Created with www.bioRender.com (accessed on 26 July 2023).

### 3. Regulation of PRAME Expression

Mainly epigenetic events regulate the expression of CTAs, such as PRAME, through mechanisms including DNA methylation of several promoter regions [11,30,31]. In fact, a correlation between hypomethylated CpG dinucleotides in TAA (tumor-associated antigen) promoters (i.e., MAGE, GAGE or PRAME) and their overexpression has been found in cancer cell lines and tissues. In CML, PRAME expression is regulated by promoter hypermethylation [11,30,32]. Another study in AML has shown 5-azacitidineinduces PRAME overexpression in blast cells from patients with no effect on CD34+ cells from healthy donors [33]. Studies have suggested that AML1-ETO and BCR-ABL fusion proteins may play a role in the upregulation of PRAME [5,34]. SOX9 has been reported to repress PRAME expression [35]; however, no correlation has been reported between expression of PRAME and SOX9 in CML patient samples [17].

In melanoma cells, PRAME expression was found to be upregulated by MZF1 in cooperation with DNA hypomethylation [36]. In another study, the downregulation of miR-211 correlated with the upregulation of PRAME mRNA/protein expression in 7 melanoma cell lines, indicating that miR-211 might be involved in the regulation of PRAME [37]. PRAME was found to act as a downstream factor of SOX17 and LIN28 in PGC, GCNIS, and seminomas, thus regulating pluripotency and suppressing somatic/germ cell differentiation [38]. In prostate cancer, PRAME was found to be a downstream target of miR-421 that inhibits PRAME expression by binding to its 3'-untranslated region (UTR) [39]. Regulated by upstream molecules, PRAME exerts its biological functions via the regulation of downstream targets in cancer [6]. In a recent study on laryngeal squamous-cell carcinoma samples, HDAC5 was identified as an upstream regulator of PRAME expression [40]. The proteins and micro-RNAs known to regulate PRAME expression at the transcription level are shown in Figure 3.

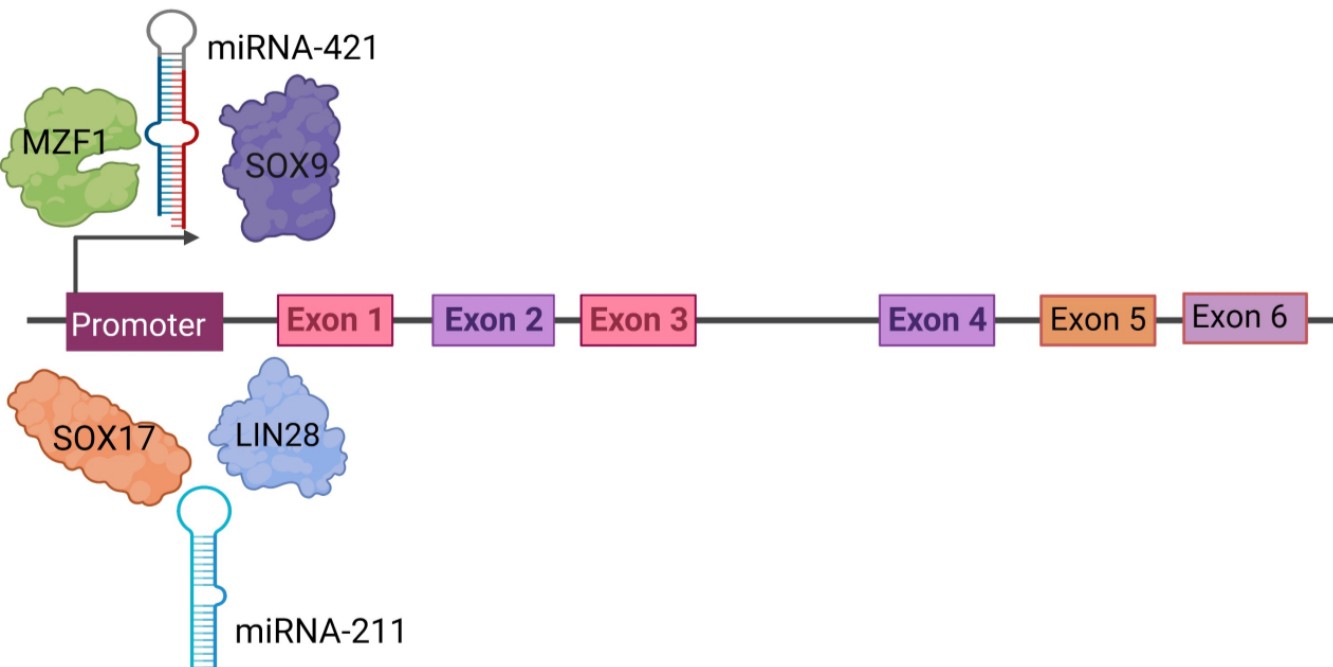

**Figure 3. Regulation of PRAME gene expression.** PRAME is regulated by upstream molecules binding to the promoter, such as SOX17, SOX9, LIN-28, MZF1, miR-211, and miR-421. The PRAME gene has at least six exons as shown in colored boxes as Exon 1 through Exon 6. Created with www.bioRender.com (accessed on 26 July 2023).

## 4. Role of PRAME in Different Cancer Hallmarks

### 4.1. Proliferation

Knockdown of PRAME expression by RNA interference in retinoic acid (RA)-resistant human melanoma was shown to restore retinoic acid receptor (RAR) signaling and resensitize cells to the antiproliferative effects of RA in vitro and in vivo [16]. PRAME knockdown was found to reduce cell proliferation, activate p53-mediated apoptosis, and increase cyclin p21 expression in vitro and in vivo in hepatocellular carcinoma (HCC) [41]. PRAME knockdown also significantly suppressed proliferation and colony formation and led to G1 cell-cycle arrest in U-2OS cells, suggesting the important role it plays in cell proliferation and disease progression in osteosarcoma [42]. A study on cervical cancer showed that PRAME promoted proliferation, migration, and invasion in cells and reduced apoptosis and G0/G1 cell-cycle arrest by activating the Wnt/β-catenin pathway [43]. A study identified PRAME target genes using CHIP-seq and found that it regulates the Cdk8 and Cdkn2d genes in ESCs after retinoic acid treatment (18). Cdk8 is a cyclin-dependent kinase that maintains embryonic stem cells and tumor cells in an undifferentiated state (19),

while Cdkn2d (cyclin-dependent kinase inhibitor 2D) is a member of the INK4 family of cyclin-dependent kinase inhibitors that generally regulate the G1-to-S phase transition (20).

On the contrary, a study revealed that the knocking down of PRAME promotes breast cancer cell proliferation and inhibits apoptosis in vitro and in vivo, indicating that PRAME might play the role of a tumor suppressor in breast cancer [44].

### 4.2. Invasion and Metastasis

Overexpression of PRAME increased the migratory and invasion potential of cervical cancer cells in vitro and in vivo [43]. PRAME was found to be aberrantly hypomethylated and activated in class 1 and class 2 uveal melanomas and associated with an increased risk of metastasis in both classes [45]. In another study, silencing of PRAME significantly reduced cell migration, without any significant effect on epithelial-to-mesenchymal transition. Knocking down PRAME did not change the protein expression of the EMT markers E-cadherin and Vimentin, or transcription factors Snail, Twist, and Zeb1. BT549 cancer cells exhibited more invasive behavior though Matrigel but not collagen I, accompanied by an increase in MMP-2 and MMP-9, in the absence of PRAME. Moreover, enlarged nuclei were observed after PRAME silencing, which is a well-known characteristic of advanced/metastatic malignancy [46]. Interestingly, one study has shown that inhibition of PRAME promotes the invasion of breast cancer cells [44]. Therefore, PRAME might have different roles in localized versus metastatic disease, thus requiring different treatment strategies for each disease setting.

PRAME silencing in head-and-neck squamous carcinoma (HNSCC) cells, followed by co-incubation with RA, decreased in vitro migration and induced apoptosis [47]. In neuroblastoma, inhibiting the cleavage of the ALK-extracellular domain led to decreased migratory potential and metastasis in vitro and in vivo, with the downregulation of several genes, including PRAME. Whether downregulation of PRAME had any causal relationship with the observation is yet to be elucidated [48].

In contrast to the above, PRAME knockdown enhanced the migration and invasion of lung cancer cells. Genes involved in cell migration, including MMP1, CTGF, CCL2, and PLAU, were upregulated in PC9 cells with PRAME knockdown. Analysis of clinical data from TCGA showed that expression of MMP1 correlated with the stage, recovery, and modality of lung cancer patients. PRAME was reported to be a tumor suppressor in lung adenocarcinoma via downregulation of E-cadherin and MMP1-mediated migration, thus leading to the prevention of EMT [49].

### 4.3. Epithelial-to-Mesenchymal Transition

PRAME acts as a tumor promoter in triple-negative breast cancer (TNBC) by increasing cancer cell motility through EMT gene reprogramming. PRAME-overexpressing cells showed an upregulation of 11 genes (SNAI1, TCF4, TWIST1, FOXC2, IL1RN, MMP2, SOX10, WNT11, MMP3, PDGFRB, and JAG1) and downregulation of 2 genes (BMP7 and TSPAN13) [50]. PRAME was found to facilitate proliferation, invasion, migration, and epithelial–mesenchymal transition of laryngeal squamous-cell carcinoma (LSCC) cells and promote tumor growth in vivo, at least partially by activating PI3K/AKT/mTOR pathways [40]. A new study showed that PRAME expression can be induced by Gas6/Axl/MAPK, which induces the expression of EMT-associated genes, cell motility, and RA-independent interaction with nuclear proteins in HCC [51].

Interestingly, silencing PRAME significantly increased bone metastasis and induced osteolytic lesions in an in vivo lung cancer model. There was a positive correlation between PRAME and E-cadherin expression supporting the notion that PRAME has a similar role in EMT as E-cadherin [49].

### 4.4. Genomic Instability

A new study in uveal melanoma has shown that PRAME is predominantly expressed in spermatogonia during meiotic cross-over in coordination with genes that lead to DNA double-strand break (DSB) repair [52]. It also showed that PRAME expression in somatic cells upregulates genes involved in meiosis, chromosome segregation, and DNA repair, induces DNA DSBs, telomere dysfunction, and aneuploidy, and alters cohesion complexes in both neoplastic and non-neoplastic cells. Uveal melanoma cells with increased PRAME expression become susceptible to PARP1/2 inhibition, suggesting an increased dependence on compensatory base excision repair pathways [52]. Therefore, PRAME also induces genomic instability as a part of its tumorigenic function.

### 4.5. Deregulating Cellular Energetics

PRAME has emerged as an important component of the retinol pathway that is known to regulate cell proliferation, differentiation, apoptosis, and vertebrate development [16,53]. Studies suggested that PRAME contributes to oncogenesis by interfering with all-trans retinol (vitamin A) pathway metabolism and its active metabolites, such as retinal, β-carotene, all-trans retinoic acid (ATRA), and 9-cis- and 13-cis-retinoic acids, together known as retinoids. Retinoids and their potential as a therapy have been investigated extensively in cancer [54,55]. Among the three isoforms of RARs, RAR-β is known to have tumor-suppressive effects in epithelial cells [10]. As mentioned before, PRAME is a dominant repressor of RAR signaling and prevents ligand-induced receptor activation upon binding to RAR in the presence of RA. Hence, cancer cells with PRAME overexpression acquire a survival advantage, enabling them to escape RA-induced cell growth arrest. Moreover, PRAME might also promote malignant differentiation of CD44+/CD24 or ALDH1A1+cancer-initiating cells (CICs) in HSNCC [13,14,27]. Hence, it is plausible that resistance to RA associated with PRAME overexpression provides an advantage not only to malignant cells, but also to precancerous cells [10,14]. Most importantly, all three of the major proteins involved in RA metabolism, including ALDH1A1, RAR-β, and PRAME, were reported to be overexpressed in HNSCC [27,47]. In addition, co-expression of ALDH1A1 and PRAME was also reported in CICs [47].

### 4.6. Apoptosis and Chemoresistance

In a study on Hodgkin's lymphoma (HL), DNA microarray analysis of cells after PRAME knockdown showed downregulation of known anti-apoptotic factors. PRAME-silenced cells had increased retinoic acid signaling with the expression of the retinoic acid metabolizing cytochrome P450 (CYP26B1), a transcriptional target of retinoic acid signaling [56]. The same study showed that 5′-azacytidine (5AC) treatment increased PRAME expression and concomitantly increased resistance of these cells to cytotoxic drugs. After knocking down PRAME in a chemo-resistant, high-PRAME-expressing cell line (L-428), an increased sensitivity to cisplatin, etoposide, and retinoic acid was observed [56]. PRAME expression was reported to confer resistance to RA-induced proliferation arrest and apoptosis by repressing expression of the endogenous RAR target genes, including RARβ and p21 [16].

When PRAME was knocked down in HCC cells, there was increased activation of p53-mediated apoptosis and increased cyclin p21 expression, with a higher proportion of cells in the G0/G1 stage. PRAME overexpression was shown to reduce apoptosis by inhibiting p-p53and Bcl2-mediated apoptosis. Studies with a tumor xenograft in nude mice also found that knockdown of PRAME inhibited tumorigenesis and vice versa [41]. Another study showed that the PRAME/EZH2 complex is able to repress TRAIL expression in a cancer-specific manner in chronic and acute myeloid leukemia [57]. When RA binds to the retinoic acid receptor (RAR), it recruits a coactivator complex to the RARE sequence in the TRAIL promoter and induces gene expression. In cancer cells, the aberrant presence of PRAME recruits EZH2 to the RAR complex, which induces trimethylation of H3K27. and epigenetically represses TRAIL gene expression [57]. Analysis of the Oncomine Research

Platform database showed that PRAME overexpression in solid tumors like breast and kidney cancers and lung and prostate melanomas and sarcomas was accompanied by a decreased expression of TRAIL [58]. In urothelial cancer, overexpression of PRAME is correlated with a poor response to chemotherapy [59].

Interestingly, a study on leukemia showed that overexpressing PRAME in KG-1 and K562 leukemia cell lines led to downregulation of S100A4 and upregulation of p53, and it significantly induced apoptosis and decreased proliferation in vitro. Upregulation of S100A4 inhibits PRAME-induced p53 upregulation. Silencing PRAME in K562 cells led to upregulation of S100A4 and downregulation of p53, and it significantly increased proliferation in vitro [60].

### 4.7. Immune Evasion

Evidence suggests that PRAME expression has implications in the regulation of the immune response. PRAME contains 21.8% (iso)leucine residues and is a leucine-rich repeat (LRR) family protein. It shares structural similarities with Toll-like receptors (TLR3, TLR4) that are widely known to play an important role in antimicrobial immune responses [9]. In addition, in leukemia cell lines, PRAME expression was found to be rapidly induced by the activation of signaling pathways related to infection and inflammation [61]. The immunogenic properties of PRAME can be used to stimulate the anti-tumor response by CD8-positive T lymphocytes.

PRAME has been investigated extensively over the last decade as a target for immunotherapy. Expression of PRAME in tumors has been shown to elicit spontaneous humoral and cellular immune responses. Vaccines and adoptive T cell therapies targeting PRAME have shown favorable safety and efficacy in inducing potent immune responses in tumors [29,62,63]. However, the role of PRAME in immune evasion has also been reported. For example, in dedifferentiated liposarcoma and leiomyosarcoma tumors, PRAME expression was found to be associated with reduced expression of antigen presentation molecules, which are considered a common mechanism of immune escape [64–67]. Furthermore, PRAME expression in dedifferentiated liposarcomas correlated with reduced expression of programmed death ligand-1 (PD-L1) [68]. This suggests that treatment with PD-L1/PD-1 inhibitors may not be beneficial in tumors that overexpress PRAME as a result of decreased target expression. Additionally, a study showed that overexpression of PRAME was associated with characteristics indicative of immune evasion, for instance, reduced numbers of antigen-presenting CD163+ and CD68+ macrophages, and increased expression of the CD47 receptor (known as the "don't eat me" receptor) on tumor cells [69]. A recent study showed that PRAME expression was found to be associated with worse survival in the TCGA breast cancer cohort, particularly in immune-unfavorable tumors. MDA-MB-468 breast cancer cells with overexpression of PRAME inhibited T cell activation and cytolytic potential, which was partly restored by silencing PRAME [70]. Additionally, PRAME knockdown reduced the expression of several immune checkpoint receptors and their ligands, including PD-1, LAG3, PD-L1, CD86, Gal-9, and VISTA [65–67,70]. PRAME knockdown induced similar levels of cancer cell killing as anti-PD-L1 atezolizumab treatment. PRAME can suppress the expression and secretion of multiple pro-inflammatory cytokines, as well as mediators of T cell activation, differentiation, and cytolysis [70].

Figure 4 attempts to summarize the role of PRAME in eight hallmarks of cancer and its potential mechanism.

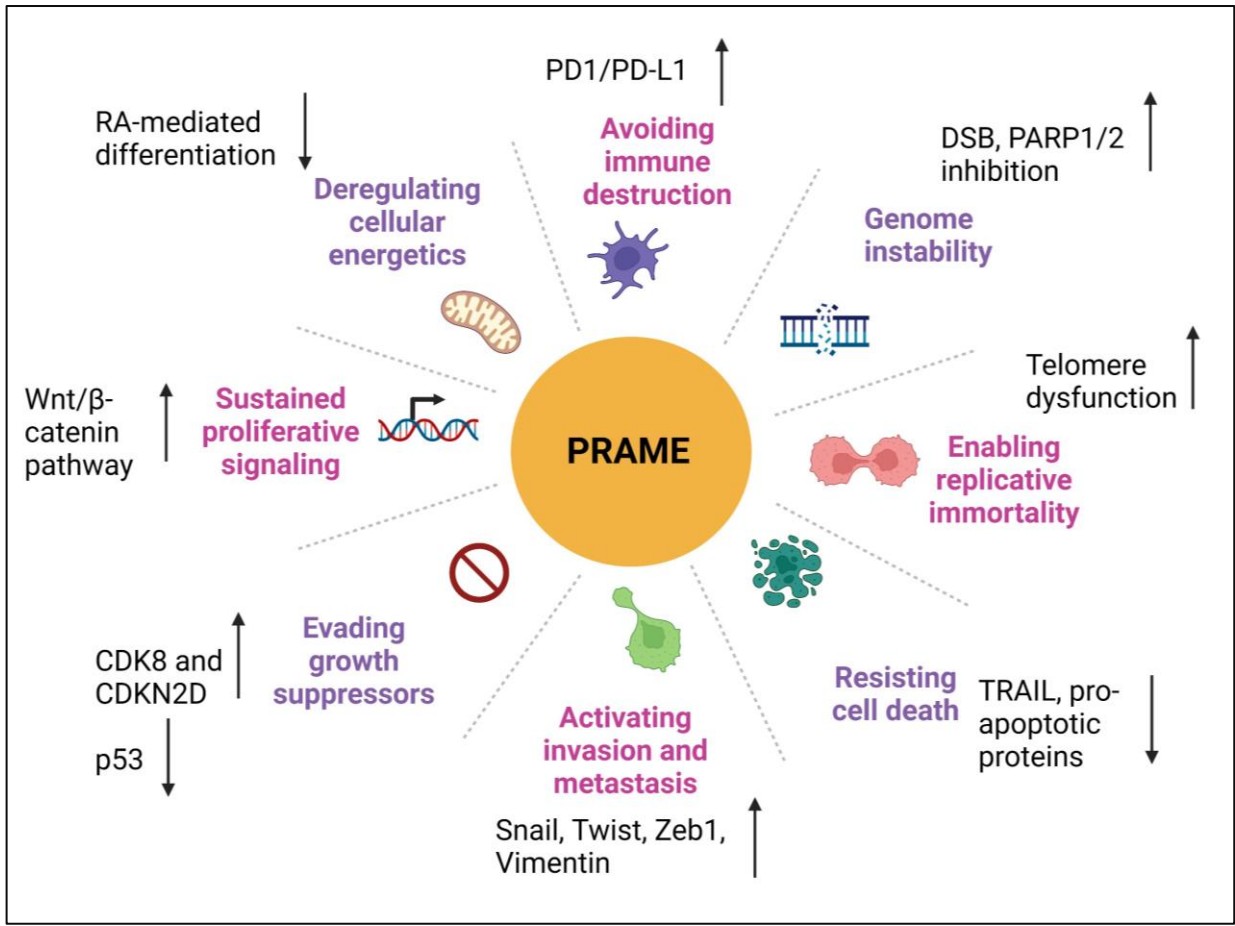

**Figure 4. PRAME plays a role in many hallmarks of cancer.** This illustration encompasses the hallmark capabilities of cancer cells in which PRAME has an active role. Upward arrows represent upregulation and downward arrows represent downregulation of the genes and processes. The hallmarks were adapted from the milestone article by Hanahan and Weinberg, "The Hallmarks of Cancer: The Next Generation" [8] Created with www.bioRender.com (accessed on 26 July 2023).

## 5. PRAME as a Biomarker

PRAME is expressed in >90% of melanomas. A study compared PRAME expression in thin metastasizing and non-metastasizing melanomas and nevi from patients. This study also found diffuse PRAME staining in >75% of epidermal and dermal melanocytic lesions and identified 58.6% of thin melanomas but did not distinguish between metastasizing and non-metastasizing melanomas [71]. PRAME expression was uniform in more than 67% in situ and invasive melanomas, whereas most severely dysplastic compound nevi (SDN) (81.0%) showed a gradient decreasing with depth. The study also found that PRAME expression is not a prognostic biomarker in melanomas ≤1.0 mm [71]. A recent study showed positive/diffuse PRAME expression (89.6%) in most malignant melanomas; however, 96.1% of nevi did not express PRAME diffusely [72]. p16 was consistently expressed in nevi (98.0%). PRAME had 89.6% sensitivity and 96.1% specificity for melanomas versus nevi. It was found to be unlikely that a PRAME+/p16− melanocytic lesion will also be a nevus, and most nevi were PRAME−/p16+. This study clearly showed that PRAME and p16 expression can be used to distinguish between melanocytic nevi and malignant melanomas [72]. A study of a Chinese cohort revealed that PRAME could be used as a marker to diagnose melanoma, and a lack of PRAME expression indicates clear cell sarcoma (CCS) in a suspected case; however, molecular confirmation of EWSR1 rearrangement is necessary for diagnosis [73]. Another study analyzed 2915 melanocytic lesions in a Polish cohort and reported that PRAME expression may be an ancillary marker to support the

diagnosis of melanoma, but the accuracy may be lower in spitzoid neoplasms [74]. Furthermore, increased PRAME expression was found to correlate with the Ki-67 proliferation index and mitotic rate; however, PRAME was not an independent prognostic marker for cutaneous melanoma. PRAME and Ki-67 were found to be useful as ancillary tools to distinguish benign from malignant melanocytic lesions [75].

A study of 103 breast tumor samples found that PRAME expression correlated significantly with unfavorable disease outcomes for patients, for both disease-free survival and overall survival times from diagnosis. Moreover, for patients who received adjuvant chemotherapy, PRAME-expressing tumors had significantly shorter relapse-free survival compared to those without PRAME expression [76]. In a study with 295 primary breast cancer patients, PRAME expression levels were found to correlate with increased rates of distant metastases and decreased overall survival, irrespective of whether they received adjuvant chemotherapy. PRAME was an independent marker of reduced metastasis-free interval in patients not having received adjuvant chemotherapy. PRAME expression was associated with tumor grade and negative estrogen receptor status. This showed that PRAME expression is a prognostic marker for clinical outcomes of breast cancer, independent of traditional clinicopathological markers [77]. Recently, an in silico study on the Cancer Genome Atlas (TCGA) and Breast Invasive Carcinoma (BRCA) dataset led to recognition of a basal-like specific gene signature composed of 11 potential unfavorable prognostic biomarkers, including PRAME [78]. PRAME was classified as a switch gene that significantly correlates with poor overall survival in patients with basal-type breast cancer. These data suggest that PRAME could serve as a prognostic biomarker and/or therapeutic target in TNBC [78].

High PRAME expression was found to be correlated with worse overall survival in medulloblastoma [79]. A study showed PRAME expression in 93% of 94 patients with primary neuroblastoma and in 100% of patients with advanced disease. PRAME expression was significantly associated with both higher tumor stage and the age of patients at diagnosis [80]. A recent study with >5800 epithelial and non-epithelial tumors showed expression of PRAME in 61% of neuroblastomas [7]. PRAME has been shown to be highly expressed in childhood acute leukemia [81], both acute myeloid leukemia (AML) and acute lymphoblastic leukemia (ALL) [82] and could be a useful marker to monitor minimal residual disease [81,83,84]. A recent study in AML showed PRAME expression in the majority of patients and it was correlated with the FAB subtype M5, cytogenetic unfavorable risk groups, and AMLs arising from myelodysplasia [85]. In another recent study, a total of 42 samples from an Indian cohort that comprised of 22 AML, 14 ALL, and 6 other forms of leukemia showed that PRAME was highly expressed in 27 (64.28%) AL patients compared to the lowest expression in healthy individuals. There was no correlation between the PRAME gene expression and clinical parameters [86].

Interestingly, various studies have suggested that increased PRAME expression can correlate with a favorable outcome and treatment response in hematological malignancies [81,87–90]. A study of 125 patients with acute promyelocytic leukemia enrolled in the Spanish PETHEMA-96 (*n* = 45) and PETHEMA-99 (*n* = 80) clinical trials had interesting results. PRAME expression in acute promyelocytic leukemia (APL) patients was significantly higher than in patients with non-M3 acute myeloid leukemia and in healthy controls [87]. Moreover, patients with high-PRAME-expressing APL had a favorable outcome. The 5-year relapse-free survival was higher in patients with >100-fold PRAME expression. PRAME levels in samples at remission had no difference with those in normal controls, while samples at relapse overexpressed PRAME. Low PRAME expression defined a subgroup of patients with a short relapse-free survival in APL, showing that PRAME could be useful as a surrogate marker [87].

Overexpression of PRAME in multiple myeloma (MM) was found to be higher in patients with an MM duration of more than 1 year and if they were pre- treated (85%) than in new cases (46.67%). PRAME expression tended to be associated with activity of LDP in blood serum [91]. Another study analyzed PRAME expression levels in MM

patients during high-dose chemotherapy, followed by auto-SCT [92]. PRAME expression was found in 68% of primary MM patients, which did not correlate with tumor mass but significantly decreased after three cycles of vincristine and doxorubicin (VAD). High PRAME expression was associated with an unfavorable prognosis [92]. In another study, PRAME overexpression was found to significantly correlate with a lower 1-year progression-free survival rate compared with low PRAME expression (20.0% vs. 88.9%) [93]. Patients with deletion of the PRAME gene had a significantly higher frequency of lambda light-chain expression than those with no deletion. Therefore, PRAME gene copy number variation (CNV) occurs in multiple myeloma, and PRAME overexpression in plasma cells might be an adverse prognostic factor for progression [93]. Another study found that both methylation and copy number variation regulate PRAME expression in MM and in patients with no homozygous deletion, PRAME expression is regulated by methylation. Overexpression of PRAME in the bone marrow was found to be an adverse prognostic factor for progression-free survival in patients treated with non-bortezomib-containing regimens [94].

PRAME was found to be overexpressed in Hodgkin's lymphoma (HL) (115/166, 69%), diffuse large B cell lymphoma (DLBCL) (104/319, 33%), follicular lymphoma (FL) (13/166, 8%), and mantle-cell lymphoma (MCL) (14/180, 8%) [95]. HL showed a significant correlation with treatment outcome, but other B cell lymphoma subtypes did not. Interestingly, Hodgkin Reed Sternberg (HRS) cells devoid of PRAME expression indicated significantly shorter overall survival and disease-specific survival. Results from this study suggested that crosstalk between CXCL13 in the microenvironment and CXCR5 on HRS cells contributes to the maintenance of tumors in PRAME-negative HL [95]. A recent study by the same group above showed that in diffuse large B cell lymphoma (DLBCL), recurrent deletion of PRAME gene was found to be associated with poor outcomes [24]. PRAME-deleted tumors were associated with cold tumor microenvironments and displayed increased cytotoxic T cell immune escape. Moreover, PRAME downregulation strongly correlated with somatic EZH2 Y641 mutations [24]. Genes regulated by PRC2 were repressed in isogenic lymphoma cell lines with PRAME-KO. PRAME was found to directly interact with EZH2 as a negative regulator [24]. Interestingly, a study on 160 patients with diffuse large B cell lymphoma (DLBCL) showed that higher PRAME expression is significantly correlated with a shorter progression-free survival (PFS) and had a trend toward shorter overall survival (OS). Patients with high PRAME expression also tended to have lower responses to chemotherapy, indicating that PRAME could serve as a prognostic biomarker for R-CHOP (rituximab plus cyclophosphamide, doxorubicin, vincristine, and prednisolone) therapy [96].

Analysis of 13 cell lines and clinical samples of esophageal squamous-cell carcinoma (ESCC) showed that PRAME is overexpressed in ESCC tissues and significantly associated with shorter disease-specific survival and hematogenous recurrence but had no correlation with overall recurrence. This indicated that PRAME is a potential biomarker for predicting hematogenous recurrence in ESCC after radical treatment and may be useful in improving the clinical outcome [97]. A recent study analyzed the oncogenic role of PRAME using 57 pairs of laryngeal squamous-cell carcinoma (LSCC) tumor tissue samples and showed that PRAME was overexpressed in the LSCC patients and correlated with the TNM staging and lymphatic metastasis [40].

A study with HCC tissues indicated that PRAME might function via DR5 RA-responsive elements and independent mechanisms and that PRAME expression is a novel prognostic marker in HCC patients [98]. In another study, PRAME expression was significantly higher in HCC tissues, compared to normal adjacent tissues, and was positively correlated with alpha fetoprotein levels and tumor size. Furthermore, PRAME expression was associated with AJCC stage and was reported to be a potential biomarker of poor prognosis for 5-year overall survival and a therapeutic target in HCC [41].

A study on Taiwanese NSCLC patients showed that PRAME expression was more frequent in squamous-cell carcinomas than in adenocarcinomas [99]. The rates of PRAME-positive tumors and its association with clinico-pathologic characteristics in East and

Southeast Asian NSCLC patients showed that it may be a promising antigen-specific immunotherapeutic target [100]. Interestingly, another study reported that PRAME is downregulated in lung adenocarcinoma and lung bone metastasis compared with normal human lungs [49]. Silencing PRAME decreased the expression of E-Cadherin and promoted proliferation, invasion, and metastasis of lung cancer cells by regulating many critical genes involved in cell migration, including MMP1, CCL2, CTGF, and PLAU. Increased survival probability was found in patients with low MMP1 expression. Therefore, in this case, PRAME was found to prevent invasion and metastasis of lung adenocarcinoma in animal models [49].

PRAME was found to be expressed in many primary and metastatic uveal melanomas and about half of the metastatic UMs co-expressed PRAME and HLA class I. Expression of PRAME was associated with clinico-pathological parameters like an increased largest basal diameter, ciliary body involvement, and amplification of chromosome 8q [101]. The 12CpG sites near the PRAME promoter were found to be aberrantly hypomethylated, and PRAME was activated in class 1 and class 2 uveal melanomas and associated with increased metastatic risk in both classes [45]. A recent retrospective case control study aimed to identify dermoscopic features that are uniquely associated with the presence of three genes associated with melanoma, including PRAME in the stratum corneum [102]. Asymmetry of color was found to be a significant predictor of PRAME expression and with other genes associated with different pigmentation patterns, the results suggested that these dermoscopic features may improve evaluation and management of pigmented skin lesions [102].

PRAME expression was found in five osteosarcoma cell lines and in more than 70% of osteosarcoma patient specimens, with high PRAME expression found to be associated with poor prognosis and lung metastasis [42]. A recent study on osteosarcoma confirmed that PRAME expression has no relation with disease evolution. However, it showed that PRAME expression is a good biomarker that may lead to detect circulating tumor cells or molecules for early diagnosis of metastasis [103]. A recent study of 350 cases showed that PRAME expression is not accurately specific but can be useful in diagnostic applications in soft tissue tumors [104].

PRAME expression was significantly increased in most epithelial ovarian carcinoma (EOC), irrespective of stage and grade, compared to normal ovaries. Interestingly, PRAME mRNA expression was associated with improved survival in the high-grade serous carcinoma (HGSC) subtype. PRAME promoter DNA hypomethylation correlated with increased PRAME expression and was very frequent in both types of ovarian cancer. PRAME protein expression showed no correlation with EOC clinicopathology or survival [89]. PRAME was reported to be associated with survival and as a potential prognostic factors for patients with stage III serous ovarian adenocarcinomas [105,106].

Microarray analysis comparing the gene expression profiles of six clinical myxoid liposarcoma samples and three normal adipose tissue samples showed upregulation of PRAME in myxoid liposarcoma. High expression of PRAME was significantly correlated with tumor diameter, the occurrence of tumor necrosis, higher histological grade, advanced clinical stage, and poor prognosis [68].

A study on head-and-neck squamous carcinoma (HNSCC) cell lines showed that PRAME was overexpressed in tumor cells but not in normal keratinocytes (HaCaT cells). PRAME overexpression in HNSCC tissue specimens correlated with the conventional parameters of poor prognosis, such as a large tumor size, high tumor grade, and lymph node involvement. In addition, PRAME was found to be overexpressed in HNSCC patients with an advanced disease (stages III and IV) [47]. Therefore, these data suggested that over-expression of PRAME in HNSCC could potentially serve as a biomarker of poor outcome and as a future therapeutic target [47]. A recent study in mucosal melanoma of the head and neck region showed that PRAME expression could be used for the accurate diagnosis of head-and-neck melanocytic tumors. Furthermore, high expression ($\geq$60%) of PRAME

was associated with specific sites (nasal cavity/nasal septum/turbinates nasopharynx, and the maxillary sinus), nodular histotype, and female sex [107].

In a panel of bladder tumors from 350 patients, 20% was found to express PRAME that also responded poorly to chemotherapy [59]. A recent meta-analysis of 14 original studies with 2421 patients showed that the PRAME expression was significantly associated with tumor stage and positive lymph node metastasis. Overexpression of PRAME positively correlated with poor disease-free survival, progression-free survival, metastasis-free survival, and overall survival [108].

Another study showed that a high PRAME expression level in follicular lymphoma patients had a negative prognostic value only in the presence of parameters determining high FLIPI-1 and FLIPI-2 risk. PRAME expression level and FLIPI-1/FLIPI-2 values together enabled the most reliable prediction of early mortality in follicular lymphoma patients [109].

A study on seminomas reported that PRAME expression was higher in seminomatous germ cell tumor of the testis (GCTT) than in nonseminomatous GCTT. Germ cell neoplasia in situ (GCNIS) and the uninvolved background testis also expressed high PRAME expression, with no differences between cases associated with seminomatous and nonseminomatous GCTT [110]. In the uninvolved background testis, PRAME expression was observed in spermatogonia (80%) and primary spermatocytes (15%), with scattered positive secondary spermatocytes and spermatids (5%). There was a strong positive correlation between age and PRAME for GCTT but not for GCNIS and the uninvolved background testis. Additionally, PRAME was not associated with dimension and pT stage, but was associated with the latter in the uninvolved background testis [110].

Table 1 lists the reports of PRAME as a biomarker in different cancers, the cohort with which the study was carried, the frequency of its expression, and its correlation with clinicopathological parameters.

**Table 1.** Diseases with PRAME as a potential biomarker.

| Cohort | PRAME Detection Frequency | Disease | Clinico-Pathological Parameters | References |
|---|---|---|---|---|
| Egyptian | 20–40% | Acute lymphoblastic leukemia | Correlated with increased overall and disease-free survival and lower relapse. | [88] |
| German | 40–60% | Acute myeloid leukemia | Positively correlated with increased overall and disease-free survival and negatively correlated to the white blood cell count at diagnosis. | [81,90] |
| Dutch, Irish | 27–53% | Breast cancer | Independent marker for poor disease-free and overall survival and distant metastases, correlates with negative estrogen receptor status. | [76,77] |
| Chinese | 80–90% | Cervical cancer | Associated with increased proliferation and migration in CC cells. | [43] |
| | 30–40% | Chronic myeloid keukemia | | |
| Canadian, Japanese | Deleted in >13% of tumors in the Canadian cohort  >30% overexpressed in the Japanese cohort | Diffuse large B cell lymphoma | Deletion of PRAME was associated with decreased overall and disease-free survival in the Canadian cohort. PRAME overexpression was correlated with shorter progression-free and overall survival, and decreased response to chemotherapy in the Japanese cohort. | [24,96] |
| Japanese | 87% tumor tissues | Esophageal cancer | Shorter disease-specific survival and hematogenous recurrence. | [97] |

**Table 1.** *Cont.*

| Cohort | PRAME Detection Frequency | Disease | Clinico-Pathological Parameters | References |
|---|---|---|---|---|
| Russian | 42–86% in lymph node, bone marrow and blood | Follicular lymphoma | Higher Ki-67 activity and larger tumor mass. Survival parameters were worse with high PRAME expression levels. Combination of both high FLIPI-1/FLIPI-2 risk and high PRAME expression level determines extremely unfavorable prognosis. | [109] |
| Polish | >75% in HNSCC tissues and lymph nodes | Head-and-neck squamous carcinoma | Correlates with the tumor grade, size, nodal involvement, and the clinical status of HNSCC patients. | [47] |
| Chinese, Japanese | 27–60% | Hepatocellular carcinoma | Correlated with high Ki-67 activity, AJCC stage, tumor size, metastasis, invasion, poor overall survival. | [41,98] |
| Turkish | 10–69% | Hodgkin's lymphoma | Correlated with shorter disease-free survival and overall survival. | [111] |
| Taiwanese, East and Southeast Asian | >30% >50% | Non-small-cell lung carcinoma | Increased in squamous-cell carcinomas compared to adenocarcinomas in all cohorts, as well as in smokers compared to non-smokers in the East and Southeast Asian cohort. No correlation with survival in the Taiwanese cohort. | [99,100] |
| Roman, Brazilian | >80% | Medulloblastoma | No significant association. | [79,112] |
| American, German | 80–90% | Melanoma | No prognostic significance in thin melanoma. PRAME+/p16- melanocytic lesion is unlikely to be a nevus but most nevi were PRAME-/p16+. | [71,72] |
| Chinese, Russian | 20–68% | Multiple myeloma | Correlated with lower progression-free survival, unfavorable prognosis. | [91–94] |
| German | >90% | Neuroblastoma | Correlated with higher tumor stage and the age of patients at diagnosis. | [80] |
| Chinese | >68% | Osteosarcoma | Associated with poor prognosis and lung metastasis. | [42,103] |
| Swedish, Norwegian, Chinese | 60–90% | Ovarian cancer | Associated with overall survival, disease-free survival, grade, stage, metastasis. | [89,105,106,113] |
| German | >40% | Renal cell carcinoma | Associated with unfavorable prognosis. | [114] |
| Italian | >70% | Testicular cancer | Associated with seminomas. | [110] |
| Danish | >20% | Urothelial cancer | Correlated with high grade and stage and poor response to chemotherapy. | [59] |

## 6. PRAME as a Target for Immunotherapy

PRAME and other CTAs have been utilized as therapeutic targets for over a decade [28]. The restricted expression of (1) PRAME in certain somatic tissues, (2) overexpression in cancer tissues, (3) immunomodulatory potential, and (4) negligible expression on immune cells make it a suitable candidate to target with immunotherapy. PRAME is overexpressed in both solid tumors and hematological malignancies [7]. Multiple pre-clinical and clinical trials have taken place over the last decade that target PRAME alone or in combination with other molecules in different cancers across the world.

One research group generated a polyclonal antibody (membrane-associated PRAME antibody 1, MPA1) against an extracellular peptide sequence of PRAME and demonstrated targeting it in vivo by radiolabeling MPA1 with zirconium-89 (89Zr-DFO-MPA1), which showed a high specific uptake in PRAME expressing hematological tumors [115]. In a study, 60 patients were treated with different doses of PRAME as an immunotherapeutic vaccine,

and no dose-limiting toxicity was reported. Adverse events included mostly injection site reactions or fever. All patients had detectable quantities of anti-PRAME antibodies after four immunizations. Patients also showed PRAME-specific CD4-positive T cells at a dose of 500 µg [116].

Despite PRAME being an intracellular antigen, novel approaches have been developed to target it using a chimeric antigen receptor (CAR) construct that encodes a targeting domain based on T cell receptor (TCR) mimic antibodies that target the peptide-HLA complex. The antibody sequence from a previously designed TCR mimic (mTCR) antibody named Pr20, which recognized the PRAME ALY peptide in the complex with HLA-A∗02 and verified expression of PRAME, was used [117]. CAR T cells (PRAME mTCRCAR T) were developed to be tested against primary samples from patients with AML and AML cell lines expressing PRAME. PRAME mTCRCAR T cells showed target-specific, HLA-mediated in vitro activity in OCI-AML2 and THP-1 cell lines, HLA-A2 cell lines expressing PRAME, and in primary AML patient samples and in vivo cell-derived xenograft models. In addition, the cytolytic activity of PRAME mTCRCAR T cells was enhanced by the treatment of the target cells with IFN-γ, as it increases PRAME expression [117,118]. Genetically modified T cells with a PRAME-specific TCR (SLL TCR T cells) were used to target medulloblastoma cells. SLL TCR T cells efficiently killed medulloblastoma HLA-A*02+ DAOY cells and primary HLA-A*02+ medulloblastoma cells. In addition, SLL TCR T cells helped control tumor growth in an orthotopic mouse model of medulloblastoma. An inducible caspase-9 (iC9) gene was introduced in frame with the SLL TCR to prevent unexpected T-cell-related toxicity, and this safety switch triggered prompt elimination of the T cells [79].

In tumors with PRAME expression, treatment approaches like PRAME-targeted immunotherapy as a vaccine (acellular PRAME vaccine, PRAME pulsed dendritic cells (DCs)), including adoptive T cell therapy, antibody therapy/chimeric antigen receptor-T cell therapy, hold great promise. In a recent study, twenty AML patients in first complete remission ineligible for allo-HSCT were treated with an autologous RNA-loaded mature dendritic cell (mDC) vaccine expressing WT1 and PRAME. This vaccine was well tolerated, with mild and transient reactions in the injection site. A total of 55% of patients remained in complete remission, while 4 of 6 relapsing patients achieved complete remission after salvage therapy and underwent allo-HSCT. The vaccine also increased five-year long-term survival to 75% [119]. Strategies that employ histone deacetylase inhibitors and demethylation agents also seem plausible. In the future, combination therapy combining PRAME vaccines or antibodies, or adoptive T cell therapy and retinoids, could be used in undifferentiated solid tumors.

A hostile tumor microenvironment (TME) poses the greatest challenge for treatment of solid tumors with immunotherapy as it adversely affects the health and persistence of T cells. In a recent study, combining a PRAME-specific T cell receptor (TCR) alongside a chimeric PD1-41BB receptor, consisting of the extracellular domain of PD-1 and the intracellular signaling domain of 4-1BB, converting an inhibitory pathway into a co-stimulatory pathway, was introduced into T cells, and this enhanced IFN-γ secretion by the CD8+ T cells, improved their cytotoxic capacity, and prevented exhaustion upon repetitive challenge with tumor cells in vitro, keeping the safety profile unaltered. In addition, a single dose of TCR-Ts co-expressing PD1-41BB was sufficient to eliminate a resistant melanoma xenograft [120].

Many articles have put forth a comprehensive review on PRAME and other CTA-based immunotherapeutic strategies in various cancers [29,121–124]. A number of immunotherapy strategies targeting PRAME are currently under clinical trials with the aim of eliciting non-toxic and long-lasting anti-tumor immune responses. Efforts in PRAME-based immunotherapies have focused on cancer vaccines and adoptive T cell therapies, as summarized in Table 2. To summarize the results of these clinical studies, PRAME could be a target for therapy for lymphoma, neuroblastoma, Wilms' tumor, and other solid tumors for which PRAME expression was found to correlate with poor prognosis across cohorts.

**Table 2.** Clinical trials with PRAME as the immunotherapy target.

| NCT Number and Name | Disease | Treatment | Start Year | Publication |
|---|---|---|---|---|
| NCT01333046, ACTAL | Hodgkin's lymphoma, non-Hodgkin's lymphoma, Hodgkin's disease | Multi TAA T cells (NY-ESO-1, MAGEA4, PRAME, Survivin and SSX), and Azacitidine | 2012 | [125] |
| NCT02203903, RESOLVE | Relapsed/refractory hematopoietic malignancies (ALL, AML, CML, MDS) | Multi TAA T cells (WT1, PRAME, and Survivin) | 2015 | [126] |
| NCT02239861, TACTASOM | Rhabdomyosarcoma | Multi TAA T cells (NY-ESO-1, MAGEA4, PRAME, Survivin, and SSX) | 2015 | NR |
| NCT02291848, TACTAM | Multiple myeloma | Multi TAA T cells (NY-ESO-1, MAGEA4, PRAME, Survivin, and SSX) | 2015 | NR |
| NCT02475707, STELLA | Leukemia, ALL | Multi TAA T cells (WT1, PRAME, and Survivin) | 2016 | NR |
| NCT02494167, ADSPAM | AML, MDS | Multi TAA T cells (WT1, NY-ESO-1, PRAME, and Survivin) | 2016 | [127] |
| NCT02743611, BP-011 | AML, MDS, uveal melanoma | BPX-701 and Rimiducid | 2017 | NR |
| NCT02789228, REST | Solid tumors (Wilms' tumor, neuroblastoma, rhabdomyosarcoma, adenocarcinoma, and esophageal cancer) | Multi TAA T cells (WT1, PRAME, and/or Survivin) | 2016 | [128] |
| NCT03093350, TACTIC | Breast cancer | Multi TAA T cells (NY-ESO-1, MAGEA4, PRAME, Survivin, and SSX2) | 2017 | [129] |
| NCT03192462, TACTOPS | Pancreatic cancer | Multi TAA T cells (NY-ESO-1, MAGEA4, PRAME, Survivin, | 2018 | [130] |
| NCT03503968, CD-TCR-001 | High risk myeloid and lymphoid neoplasms | MDG1011 | 2018 | NR |
| NCT03652545, REMIND | Brain tumor TAA, | Multi TAA T cells (WT1, PRAME, and/or Survivin) | 2018 | NR |
| NCT03686124 | Solid tumor | Autologous PRAME-targeting TCR-engineered T cells ACTengine® IMA203/IMA203CD8 as monotherapy or in combination with nivolumab | 2019 | NR |

## 7. Conclusions and Future Perspectives

A number of CTAs had made it to the priority-ranked list of cancer vaccine target antigens based on predefined and pre-weighted objective criteria in a study conducted by the National Cancer Institute in 2009 [131]. A total of 75 representative antigens were selected for comparison and ranking based on their (i) therapeutic function; (ii) immunogenicity; (iii) role in oncogenesis; (iv) specificity; (v) expression level and percent of antigen-positive cells; (vi) stem cell expression; (vii) number of patients with antigen-positive cancers; (viii) number of antigenic epitopes; and (ix) cellular location of antigen expression. Although, none of the 75 antigens satisfied all of the parameters required to be an ideal cancer antigen, 46 were immunogenic in clinical trials and 20 of them had suggestive clinical efficacy in the category of "therapeutic function". PRAME did not make it to the list; however, MAGEA3, NY-ESO-1, MAGEA1, OY-TES1, SP17, XAGE1, and PAGE-4 came up in the top antigen's list [131], reinforcing the significance of CTAs as a therapeutic target

in various cancers. PRAME is also an important tumor antigen with different oncogenic functions, and therefore targeting it holds promise as a therapy.

The function of PRAME in tumor progression is complex and context-dependent [24]. In most cases, PRAME expression drives tumorigenesis, and in some cases, PRAME overexpression can reduce malignancy [22]. This kind of dual nature of a cancer antigen is not new and has been seen before in a few other genes, such as MUC1 and TGF-β [132–134], in which they act as either tumor suppressors or promoters in a context- and tissue-dependent manner. There is controversy over the function of PRAME in leukemic cells due to conflicting reports in the literature. Numerous studies on solid malignancies have reported that high PRAME expression correlates with advanced-stage disease and poor survival, whereas in pediatric acute leukemia, overexpression of PRAME was reported to predict good outcomes [81,88,90]. Therefore, age might be one of the parameters that affect PRAME function in leukemia. Therefore, further molecular characterization of the cells with age as a variable is needed to conclude whether PRAME can be a good biomarker of prognosis. The function of PRAME varies according to different cell lineages and depends on the different genetic or epigenetic mechanisms of the specific tumor tissue. Due to its paradoxical role of both promoting and inhibiting tumorigenesis, direct inhibition of PRAME function may not prove to be therapeutically beneficial eventually. Therefore, it is important to classify tumor tissues, subtypes, stages, and genetic signatures in a clinical setting, in which PRAME functions solely as an oncogene and can be targeted for a better outcome. It has also been seen in multiple studies that the expression of PRAME and its correlation with clinico-pathological features varies with cohorts worldwide. Therefore, cohort comparison is also important in developing a PRAME-based anti-tumor therapy on a large scale.

PRAME makes an attractive target for cancer therapy due to the following main reasons:

(1) Its restricted expression in the testis, ovaries, and endometrium and overexpression in a number of cancer tissues, including 80–90% of primary and metastatic melanoma [9], 27–53% breast cancers [53], >90% neuroblastomas [53,80], 40–60% acute myeloid leukemia (AML) [82,90], 20–40% acute lymphoblastic leukemia (ALL) [9,82], 20–50% myeloma [53], and 30–40% chronic myeloid leukemia (CML) [53,135];

(2) It has the ability to elicit T-cell-mediated immune response.

PRAME expression was also found in the stem cells of CML [136], indicating that targeting PRAME could abolish the cancer-initiating, resistant cell population, although this needs to be investigated in other cancers. Monoclonal antibodies (mAbs) have shown anti-tumor efficacy in the clinic against various cancers [137,138]. However, a major drawback of currently available mAbs is that they exclusively bind to cell-surface and extracellular antigens, while many of the aberrantly expressed proteins in cancer, including PRAME, are intracellular.

PRAME being an intracellular membrane protein [9,22,61] makes it impossible to target using traditional antibodies that are usually directed at cell-surface antigens, and currently it cannot be inhibited using small molecules, thus making it undruggable. Therefore, most studies with PRAME-based immunotherapy have focused on utilizing circulating PRAME-specific T cells and engineering of antigen-specific T cell receptor (TCR) T cells [29]. Nevertheless, a number of research groups have investigated the efficacy of different strategies to target PRAME. Following proteasomal processing, the PRAME peptide ALYVDSLFFL (ALY) (amino acid 300–309) is presented as HLA-A*02:01 molecules for recognition by the TCR on the surface of cytotoxic T cells [23,139,140]. In one study, a fucosylated Fc form (Pr20M) of a TCR mimic (TCRm) human IgG1 antibody called Pr20, which recognizes the cell-surface ALY peptide/HLA-A2 complex and binds to PRAME+HLA-A2+ cancers, was found to trigger antibody-dependent cellular cytotoxicity (ADCC) against PRAME+HLA-A2+ leukemia cells and was therapeutically effective against xenograft models of human leukemia in mice. In some tumors, IFN-γ treatment significantly increased Pr20 binding, mediated by induction of the immunoproteasome catalytic subunit

β5i [139]. Several groups have demonstrated the ability to generate ALY/HLA-A2-specific CD8+ cytotoxic T lymphocytes (CTLs) that can specifically lyse PRAME+HLA-A2+ tumor cells and are reactive against primary leukemia [141–143], suggesting that this epitope is presented and can be a promising target for immunotherapy. Clinical trials also provide evidence that patients vaccinated against PRAME can develop PRAME-specific CTLs [144] and helper T cells [145]. However, there are several challenges to develop cellular and vaccine-based strategies. CTL-based therapies are patient-specific and often require rigorous manipulation before reinfusion, while vaccines may not be effective and responses are unpredictable, depending on the patient's immune status [146]. Additionally, the immunoproteasome generally favors cleavage after hydrophobic residues, which generates peptides that can fit into the groove of HLA-I [147,148]. Several antigens are restricted to a specific proteasome form, and more knowledge on this can help to determine promising immunotherapy strategies against these targets [149].

Targeting PRAME with immunotherapy in combination with ATRA was also shown to be promising in retinoid-resistant oral carcinoma [121]. In addition, immunotherapy using circulating antigen-specific T cells and engineered T cells expanded ex vivo has gathered attention since it does not directly rely on the patient to actively generate a huge repertoire of antigen-specific immune cells and has been shown to elicit durable response rates across malignancies [150].

The use of PROteolysis TArgeting Chimeras (PROTACs) to degrade oncogenic proteins has emerged as a potential therapeutic strategy over the last decade [151]. PROTACs are heterobifunctional molecules that comprise of one ligand that binds to a protein of interest (POI) and another that binds to an E3 ubiquitin (E3) ligase, connected via a linker. PROTACs recruit the E3 ligase to the POI and lead to proximity-induced ubiquitination and degradation of the POI by the ubiquitin proteasome system (UPS). PROTACs have been developed efficiently to degrade a wide range of cancer targets against a variety of tumor types [151]. Therefore, PROTACS could be developed using PRAME-targeted antibody sequences as a promising strategy to eliminate the oncogenic properties of the latter. However, in immunologically active tumors with overexpression of PRAME, degrading it might be a trade-off with limiting immune infiltration into the tumor microenvironment. On the contrary, immunologically cold tumors with PRAME overexpression may have a better outcome on PRAME degradation, where having PRAME as an immunological target was intrinsically not useful. Therefore, it is imperative to study the expression pattern and precise role of PRAME and its interaction with the microenvironment in different tumors in order to develop targeted therapies.

**Funding:** This research received no external funding.

**Institutional Review Board Statement:** Not applicable.

**Informed Consent Statement:** Not applicable.

**Data Availability Statement:** Not applicable.

**Conflicts of Interest:** The author declares no conflict of interest.

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
