# Peer review of "Preferentially Expressed Antigen in Melanoma Is a Multifaceted Cancer Testis Antigen with Diverse Roles as a Biomarker and Therapeutic Target"

_2673-8937, doi:10.3390/ijtm3030024_

Round 1

Reviewer 1 Report

References journal are usually cited through their own abbraviations. In this long paper you could report the current clinical advantage of this marker, and its impact on the diagnosis.  Your research is largely extended, but referred without a scale of importance. 

Author Response

Thank you for your comments. I have updated the manuscript with the current clinical advantage of this marker and stated its importance in the Discussion section.

Reviewer 2 Report

The author described the various features of PRAME in light of 'cancer hallmarks' and also discussed its possible role as a biomarker or a therapeutic target. However, some points would be clarified. 

1. 'it encodes the human leucocyte antigen (HLA)-A24' should be 'it encodes the epitope presented by human leucocyte antigen (HLA)-A24' in line 31.

2. The description of PRAME's role in protein degradation is complicated (line 128-137). Alternation of a part of Figure 2 would be recommended.

3. 'PRAME as a biomarker' in line 328 would be '4. PRAME as a biomarker'.

4. 'PRAME as a target for immunotherapy' in line 507 would be '5. PRAME as a target for immunotherapy'

5. The paper below should be cited in addition to ref.81 and 83 in line 373.

Matsushita M, Ikeda H, et al. 'Quantitative monitoring of the PRAME gene for the detection of minimal residual disease in leukaemia. Br J Haematol. 2001 Mar;112(4):916-26. PMID: 11298586.

6.The reason why increased PRAME expression in leukemia can correlated with good prognosis would be discussed considering the function of PRAME in leukemia.

7. As the author pointed out, the function of PRAME varies depending on the type of cancer, thus, the author might be able to suggest that which type of cancer is especially suitable or unsuitable for PRAME-targeting therapy and which would be discussed with the results of clinical trials.

Author Response

Thank you so much for your valuable comments. Please see my responses below:

  1. 'it encodes the human leucocyte antigen (HLA)-A24' should be 'it encodes the epitope presented by human leucocyte antigen (HLA)-A24' in line 31. This sentence has been corrected in revised manuscript in lines 29-31.
  1. The description of PRAME's role in protein degradation is complicated (line 128-137). Alternation of a part of Figure 2 would be recommended.

The whole paragraph on PRAME's role in protein degradation was modified in page 5. However, figure 2 is the most simplistic version that shows PRAME as a part of the E3-ligase complex that ubiquitinates the target protein. Therefore, the figure has not been altered but the figure legend has been updated in page 6.

  1. 'PRAME as a biomarker' in line 328 would be '4. PRAME as a biomarker'.

This has been corrected in the revised manuscript.

  1. 'PRAME as a target for immunotherapy' in line 507 would be '5. PRAME as a target for immunotherapy'

This has been corrected in the revised manuscript.

  1. The paper below should be cited in addition to ref.81 and 83 in line 373.

Matsushita M, Ikeda H, et al. 'Quantitative monitoring of the PRAME gene for the detection of minimal residual disease in leukaemia. Br J Haematol. 2001 Mar;112(4):916-26. PMID: 11298586.

The reference has been added in line 373 in the revised manuscript.

6.The reason why increased PRAME expression in leukemia can correlated with good prognosis would be discussed considering the function of PRAME in leukemia.

PRAME expression has been shown to have oncogenic effects in leukemia in many studies. However, a couple of studies have shown that PRAME expression led to decrease in proliferation of leukemia cell lines. Therefore, there is controversy regarding the function of PRAME in leukemia due to contrasting reports and whether increased PRAME expression can be a marker of good prognosis remains elusive. The function of PRAME changes with cell type and origin, cohort and expression of other molecular markers. In pediatric leukemia, it has been shown that PRAME overexpression predicts good outcome. Therefore, further research is required to determine the role of PRAME with increased certainty. I have mentioned this is the Discussion in lines 585-599.

  1. As the author pointed out, the function of PRAME varies depending on the type of cancer, thus, the author might be able to suggest that which type of cancer is especially suitable or unsuitable for PRAME-targeting therapy and which would be discussed with the results of clinical trials.

To summarize the clinical trial data so far, I have mentioned the type of cancers that might benefit from PRAME-targeted therapy in lines 564-567 and in the Discussion section in the revised manuscript.

Round 2

Reviewer 1 Report

The text has been ameliorated.